# SMORE-DRL: Scalable Multi-Objective Robust and Efficient Deep Reinforcement Learning for Molecular Optimization

## Abstract

The adoption of machine learning techniques within the domain of drug design provides an opportunity of systematic and efficient exploration of the vast chemical search space. In recent years, advancements in this domain have been achieved through the application of deep reinforcement learning (DRL) frameworks. However, the scalability and performance of existing methodologies are constrained by prolonged training periods and inefficient sample data utilization. Furthermore, generalization capabilities of these models have not been fully investigated. To overcome these limitations, we take a multi-objective optimization perspective and introduce SMORE-DRL, a fragment and transformer-based multi-objective DRL architecture for the optimization of molecules across multiple pharmacological properties, including binding affinity to a cancer protein target. Our approach involves pretraining a transformer-encoder model on molecules encoded by a novel hybrid fragment-SMILES representation method. Finetuning is performed through a novel gradient-alignment-based DRL, where lead molecules are optimized by selecting and replacing their fragments with alternatives from a fragment dictionary, ultimately resulting in more desirable drug candidates. Our findings indicate that SMORE-DRL is superior to current DRL models for lead optimization in terms of quality, efficiency, scalability, and robustness. Furthermore, SMORE-DRL demonstrates the capability of generalizing its optimization process to lead molecules that are not present during the pretraining or fine-tuning phases.

## 1 Introduction

Successfully developing a drug is a tremendously time-consuming, expensive and difficult endeavour. On average, it takes 10-15 years and costs \$1–2 billion USD to deliver a new drug to market (Sun et al., 2022). The objective of drug design is to identify molecules that exhibit multiple pharmacological properties characteristic of pharmaceutical-grade drugs, ensuring they are safe and efficacious. Drug design thus can be modelled as a multi-objective optimization (MOO) problem. One of the main challenges of drug design is effectively navigating the immense chemical search space, which is estimated to contain $10^{20}$ and $10^{200}$ possible drug-like molecules (Brown, 2015).

Machine learning methods, including deep reinforcement learning (DRL), offer a promising solution to this problem (Al-Jumaily et al., 2023; Goel et al., 2021; Gottipati et al., 2021; Kim et al., 2021; Pereira et al., 2021; Popova et al., 2018; Ståhl et al., 2019; Tang et al., 2023; Wang & Zhu, 2024; Yang et al., 2021). Molecular optimization is a process that requires a model to perform minor modifications on a lead molecule to improve its drug-like qualities while preserving structural similarity. As molecules with comparable structures are anticipated to exhibit similar behaviours, this approach aims at preventing the model from producing unrealistic or undesirable molecules. This differs from molecular generation, where a model's task is to generate novel and diverse compounds from scratch (Ståhl et al., 2019). Additionally, present methodologies are hindered by lengthy training requirements and sub-optimal use of training data, resulting in impaired scalability and performance.

In this work, we present SMORE-DRL (**S**calable **M**ulti-**O**bjective **R**obust and **E**fficient **D**eep **R**einforcement **L**earning), a gradient-alignment-based multi-objective DRL (MODRL) framework for molecular optimization. The key contributions of this research include: (1) molecular optimization is modelled naturally as a Pareto-based multi-objective reinforcement learning problem where the challenges of gradient dominance and conflict are addressed with gradient alignment inspired by a study from multi-task learning; (2) a novel molecular tokenization strategy is proposed to represent the a molecule as a hybrid of fragments and SMILES, enabling efficient policy learning and effective representation of any new molecules; and (3) a synergistic integration of gradient alignment, hybrid fragment-SMILES representation, contrastive learning, and a transformer-encoder allows for scalability and generalization capability superior to existing MODRL methods. Moreover, SMORE-DRL demonstrated its ability to effectively scale and generalize its optimization process to new molecules after fine-tuning. This is a particularly notable aspect of our work, as existing DRL methods lack scalability and their generalization capacities are under-explored in current literature.

## 2 Related Work

The fundamental techniques of MODRL and aligned multi-task learning, closely related to this study, are reviewed below. In addition, see Appendix A.1 for a review of transformer-encoder architectures and MLM.

### 2.1 Deep Reinforcement Learning

Reinforcement learning (RL) agents learn through a trial-and-error process guided by the Markov decision process (MDP). See Appendix A.2 for an introduction to basic RL concepts. When the RL task entails exploring a vast state or action space, as is often the case in drug design, learning an exact optimal policy or value function can become computationally intractable. Thus, DRL is used to approximate policies or value functions (Arulkumaran et al., 2017). The actor-critic framework approximates both and has been leveraged by various drug development frameworks (Al-Jumaily et al., 2023; Goel et al., 2021; Gottipati et al., 2021; Pereira et al., 2021; Popova et al., 2018; Ståhl et al., 2019; Tang et al., 2023; Wang & Zhu, 2024; Yang et al., 2021). The actor model is responsible for learning a parameterized policy $\pi_{\theta_A}$. This is guided by feedback known as temporal difference (TD) error from the critic model, which evaluates the actor's actions based on the state. One approach to this is by learning the advantage function $A^\pi(s,a) = Q^\pi(s,a) - V^\pi(s)$, which measures the desirability of taking action $a$ compared to alternative actions available from state $s$ (Graesser & Keng, 2019).

### 2.2 Multi-Objective Deep Reinforcement Learning

MODRL is a domain within machine learning and also a family within MOO focused on simultaneously optimizing two or more objectives (Liu et al., 2015). In an MODRL setting, the reward function is extended to a vector of size $K$, which represents $K$ different objectives (Nguyen et al., 2020). MODRL approaches have been developed for molecular design. Zhou et al. (2019) developed Molecule Deep Q-Networks (MolDQN), a multi-objective molecular optimization framework that implements double deep Q-learning (DDQN) and randomized value functions. For the optimization task, an episode starts with a seed lead molecule and in each timestep, MolDQN optimizes the molecule through one of the following actions: (1) atom addition, (2) bond addition, and (3) bond removal. A *linear weighted sum method* is used for MOO. Deep Fragment-based Multi-Parameter Optimization (DeepFMPO), introduced by Ståhl et al. (2019), is an actor-critic multi-objective method for molecular optimization. In this work, a library of fragments is derived from a set of lead molecules and fragments are encoded using a balanced binary tree such that similar molecules have similar binary encoding. One modification step involves replacing a fragment in the lead molecule with a similar fragment from the fragment library. A constrained reward function is used, where a molecule is either assigned a constant positive reward for each objective achieved or a reward of zero. If all objectives are met, the reward is doubled. A dynamic reward mechanism is also implemented, where the model is penalized if it begins to under-

perform compared to previous epochs. Bolcato et al. (2022) expand DeepFMPO to include 3D-shape and electrostatics in the similarity measurements. This extension was applied because a seemingly minor alteration to a SMILES string can significantly impact its 3D structure. Consequently, the revised representation of fragments is suggested to achieve a more precise similarity measure. When recognizing the existence of other RL approaches for molecular optimization, the three aforementioned MODRL approaches representatively form the benchmarks to compare with our proposed framework.

### 2.3 Aligned Multi-Task Learning

Two potential issues that arise when solving an MOO problem directly using gradient descent are dominating and conflicting gradients. A dominating objective gradient is characterized by the largest magnitude, which leads to a bias in the solution favouring the corresponding task (Senushkin et al., 2023). When two objective gradients are conflicting, an increase in the solution towards one objective decreases the solution for the conflicting objective. Conflicting gradients are characterized by having a negative cosine similarity (Yu et al., 2020). To address these challenges in the context of multi-task learning, Senushkin et al. (2023) propose aligned-multi-task learning (AMTL). Let $\mathcal{L}_k(\boldsymbol{\theta})$ represent the objective of task $k$, where there are $K > 1$ tasks that are associated with a set of model parameters $\boldsymbol{\theta}$. The training objective is to converge to a set of $\boldsymbol{\theta}^*$ defined as follows:

$$\boldsymbol{\theta}^* = \arg\min_{\theta \in \mathbb{R}^m} \left\{ \mathcal{L}_0(\boldsymbol{\theta}) \overset{\text{def}}{=} \sum_{k=1}^{K} \frac{1}{K} \mathcal{L}_k(\boldsymbol{\theta}) \right\}. \tag{1}$$

To mitigate conflicting and dominating gradients, AMTL aligns the principal components of an initial linear system of gradients. This process can be interpreted as re-scaling the axes of a coordinate system that is determined by the principal components, such that the minimal singular value of the gradient matrices is identified and all other singular values are adjusted to match it, resulting in aligned gradients. Subsequently, these aligned gradients are combined into a common gradient (Senushkin et al., 2023). Inspired by AMTL which formulates a multi-task learning problem to an MOO problem, we integrate the same gradient alignment technique to solve MODRL for drug design in this study. A detailed description of AMTL-based MODRL can be found in Appendix A.3.

## 3 Methods

In this section, we discuss the data pre-processing, pretraining, and fine-tuning processes carried out in our SMORE-DRL framework.

### 3.1 Data Preparation: Fragments-SMILES Hybrid Tokenization Strategy

The dataset used for transformer-encoder pretraining is the MolGen task of the Therapeutics Data Commons (TDC) (Huang et al., 2021), a set of the ChEMBL, MOSES, and ZINC-250K datasets. We canonicalized all SMILES strings, and only kept strings with a maximum of 100 characters, resulting in a pretraining dataset of 4 million molecules. We then fragmented each molecule using the fragmentation method from HierVAE by Jin et al. (2020), which breaks single bonds extending from ring atoms (Ståhl et al., 2019). This method is also used by DeepFMPO (Ståhl et al., 2019) and DeepFMPOv3D (Bolcato et al., 2022).

While fragmentation is a good technique for reducing the chemical search space, it may result in a vast token dictionary size. Figure 4 in Appendix A.4 is a fragment frequency chart based on the 4-million molecule dataset, which shows that nearly 68,000 fragments were extracted. Most of these fragments are rarely encountered in the dataset, with 95% appearing fewer than 100 times. Moreover, building a token dictionary solely from the pretraining dataset will create obstacles during fine-tuning tasks. Given the sparse nature of fragment occurrences, it is likely that molecules used for fine-tuning will contain fragments not present in the dictionary, especially if the fine-tuning dataset differs from the one used for pretraining.

To reduce the dictionary size while still representing fragments that are absent or infrequently encountered, we propose a novel hybrid tokenization strategy that uses both fragments and SMILES. Following the construction of a fragment dictionary from the pretraining dataset, we append tokens for SMILES and exclude all fragments that appear less than twice. This reduces the dictionary size from 68,000 to approximately 41,130 tokens. As a result, if a molecule contains a fragment not found in the reduced dictionary, that fragment is represented atom by atom. See Appendix A.4 for a diagram of the hybrid tokenization strategy. As part of our ablation studies in Section 4.2, we demonstrate that further reducing the token dictionary by retaining only the most frequently encountered fragments (resulting in molecules being primarily represented by SMILES atoms) hinders training performance.

## 3.2 Pretraining

SMORE-DRL utilizes a transformer-encoder model inspired by architectural aspects of the Bidirectional Encoder Representations from Transformers (BERT) model introduced in MTL-BERT by Zhang et al. (2022), a multi-task learning model pretrained on SMILES strings and fine-tuned for downstream ADMET tasks. Rather than representing molecules by SMILES atom tokens as was done in MTL-BERT, we adopt our hybrid fragment token representation. SMORE-DRL employs a combination of two pretraining tasks: MLM and contrastive learning.

### 3.2.1 Masked Language Model (MLM)

Given a training batch of molecules, they are fragmented and encoded into their token representations. Unlike the static masking technique used for the MLM in the original BERT model, where sequences are masked once and reused throughout training, we employ the dynamic approach introduced by Liu et al. (2019) in Robustly Optimized BERT Pretraining Approach (RoBERTa). In each training batch, 20% of the tokens are randomly selected for masking, with 90% of those selected tokens end up being masked. If a selected token is part of a sequence of atom tokens representing a fragment that does not exist in the token dictionary, 20% of that fragment's atom token sequence is also masked. The masked molecule token sequence is then passed into the encoder model, which attempts to accurately reconstruct the original values of the masked tokens. See Appendix A.5 for a diagram of the MLM process.

### 3.2.2 Contrastive Learning

We further refine the SMORE-DRL's contextual understanding of molecules by allowing it to align its representations of similar molecules. This is particularly valuable during fine-tuning, where the model is tasked with optimizing lead molecules over multiple timesteps while ensuring that the optimized molecules in the earlier timesteps retain chemical similarity to the original lead molecule. To accomplish this, we introduce a straightforward contrastive learning technique that builds upon the MLM approach. Rather than directly masking tokens in the fragment sequence as previously described, a "separation" token is inserted at the end of the sequence, followed by an augmented version of that sequence. Augmentation involves randomly selecting a token to replace with a fragment token from the token dictionary. If the selected token belongs to a sequence of atom tokens representing a fragment that is not in the token dictionary, the entire SMILES atom sequence for that fragment is replaced with a randomly selected fragment token. The masking process consists of keeping the original molecule sequence fully visible to the model, while masking 25% of the augmented sequence using the same technique described earlier. See Appendix A.5 for a diagram of the contrastive learning process.

## 3.3 Fine-Tuning

For the fine-tuning phase, SMORE-DRL utilizes a novel multi-objective actor-critic framework with three pretrained encoder models: a masker, an actor, and a critic.

### 3.3.1 Agents

**Masker Model:** The masker model, denoted as $\pi_{\theta_M}$, is responsible for selecting which tokens to mask from the lead molecule token sequence. At each timestep, it masks at least one token and up to 70% of the token sequence. The model is designed to prefer masking fragment tokens over SMILES atom tokens. The loss for the masker model is:

$$L\left(\theta_M\right) = \frac{1}{T}\sum_{t=0}^{T}\sum_{k=0}^{K}\left(-\hat{A}_k^{\pi_{\theta_A}}\left(s_t, a_t\right)\log\pi_{\theta_M}\left(a_t \mid s_t\right)\right). \tag{2}$$

**Actor Model:** After the lead molecule token sequence is masked by the masker model, it is passed to the actor model, denoted as $\pi_{\theta_A}$. The actor model utilizes the same training head that was used during the pretraining phase. Hence, its task is to replace the masked tokens with tokens from the token dictionary. However, rather than focusing on recovering the original tokens, the actor's task is to replace the masked tokens with new tokens so that the resulting sequence represents an optimized yet chemically similar version of the lead molecule. The actor model employs AMTL (Senushkin et al., 2023) to identify a common objective gradient, thereby avoiding conflicting and dominating gradients, which ensures that all molecular properties are optimized equally. This process involves obtaining a gradient matrix that collects all $K$ objective gradients, represented as $\boldsymbol{G} = \{\boldsymbol{g}_1, \cdots, \boldsymbol{g}_K\}$, where $\boldsymbol{g}_k = \nabla L_k\left(\theta_A\right)$ and

$$L_k\left(\theta_A\right) = \frac{1}{T}\sum_{t=0}^{T}\left(-\hat{A}_k^{\pi_{\theta_A}}\left(s_t, a_t\right)\log\pi_{\theta_A}\left(a_t \mid s_t\right)\right). \tag{3}$$

$\boldsymbol{G}$ is then processed into the gradient matrix alignment algorithm to compute a common gradient, which is used to update the model. Our AMTL-based actor model optimization algorithm is given in Algorithm 1 of Appendix A.3. It is crucial for the masker and actor to work in tandem. If the actor performs well, this will be reflected in the the masker's loss, as the masker utilizes the advantage function derived from the actor model's policy. The fine-tuning process is illustrated in Figure 1.

**Critic Model:** The optimized token sequence is then fed into the critic model, $V_{\theta_C}$, which generates a vector of size $K$, corresponding to $K$ properties. The critic's output reflects its assessment of the desirability of sequence's desirability as a potential drug candidate. The loss of the critic model is:

$$L\left(\theta_C\right) = \frac{1}{T}\sum_{t=0}^{T}\sum_{k=0}^{K}\left(r_{t,k} + \hat{V}_{\theta_C k}^{\pi_{\theta_A}}\left(s_{t+1}\right) - V_{\theta_C k}^{\pi_{\theta_A}}\left(s_t\right)\right)^2. \tag{4}$$

### 3.3.2 Reward System

To assess a molecule's potential as a drug candidate, we use the following three properties: (1) logarithm of partition coefficient (ClogP), which impacts a drug's administration, absorption, transport and excretion, (2) synthetic accessibility score (SAS), which measures the difficulty of synthesizing a molecule, and (3) binding affinity score (BAS) to LPA1, which quantifies the binding capability of a drug to a target protein (Brown, 2015; Ertl & Schuffenhauer, 2009; Li et al., 2019). However, the number and type of properties can be tailored to any specific optimization task. RDKit is used for ClogP and SAS calculations, and QuickVina2-GPU-2.1 (Tang et al., 2024) was used to calculate BAS. Lysophosphatidic acid receptor 1 (LPA1/LPAR1), a bioactive lipid mediator primarily derived from membrane phospholipids, is chosen as the target protein for BAS. LPA Receptors (LPARs) have been found to be over-expressed in multiple types of cancer, with LPA1 specifically expressed in ovarian cancer, breast cancer, liver cancer, gastric cancer, pancreatic cancer, lung cancer, glioblastoma and osteosarcoma. LPA1 promotes metastasis and tumor motility, making it a natural choice for targeting in efforts to inhibit cancer spread and cell movement (Lin et al., 2021).

To convert a property value to a reward, we treat all properties to be minimized and normalize property values. The reward for molecule $m$, where property $p$ is ClogP or SAS, is

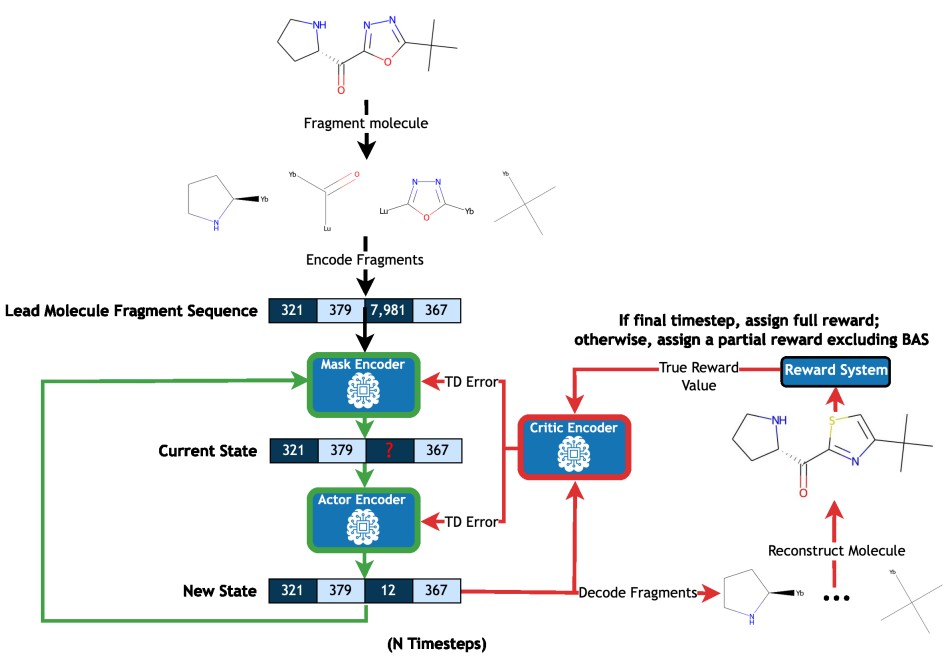

Figure 1: Our MODRL fine-tuning process for one molecule.

defined as:

$$r_{p,m} = \frac{p_{thresh} - p_m}{p_{thresh} - p_{true\_min}}, \tag{5}$$

where $p_{thresh}$ is the target maximum parameter that is set for $p$, $p_m$ is the $p$ score for molecule $m$, and $p_{true\_min}$ is the true minimum of $p$. The $p_{thresh}$ parameter controls the difficulty of the optimization task for that property, where a lower threshold that optimized molecules must achieve. When $p$ is BAS, the absolute mean BAS score of the lead molecule set ($|p_{lead\_mean}|$) is used, as no true minimum exists for BAS:

$$r_{p,m} = \frac{p_{thresh} - p_m}{|p_{lead\_mean}|}. \tag{6}$$

A widely known challenge of utilizing molecular docking is that it requires significant time (Thafar et al., 2022). To overcome this, for all intermediate, we assign a partial reward to the molecule, and in the final timestep, BAS is calculated and a full reward is given.

The following three criteria are also examined for each epoch: (1) validity: the ratio of chemically valid optimized molecules, checked using RDKit, (2) novelty: the ratio of optimized molecules that are different from the lead molecules from which they were derived, and (3) uniqueness: the ratio of unique molecules in the optimized molecules (Mukaidaisi et al., 2022). In each timestep, if a molecule is valid, unique, and novel, it is assigned its reward, else it is assigned a reward of -1 for each objective, for a total reward of -3 if it is the final timestep. If all properties are achieved by a molecule, it is provided with extra reinforcement by doubling the final reward.

Thus, the output of the reward function is $\mathbf{r}^K = \left[r^1, r^2, \dots, r^K\right]$, where $K = 3$ for the final timestep and $K = 2$ for all intermediate timesteps. The values of the reward system are listed in Table 1 of the next section.

## 4 EXPERIMENTS & RESULTS

### 4.1 PRETRAINING

The encoder model was pretrained for six consecutive epochs on a combination of MLM and contrastive learning tasks, where the same training head was used throughout learning.

While the main experiment employs the 2-frequency token dictionary (68,000 tokens), we also investigate the effect that different dictionaries have on pretraining and fine-tuning. The pretraining results using the 2-frequency, 100-frequency (3,460 tokens), and 1000-frequency (790 tokens) token dictionaries can be found in Appendix A.6.

## 4.2 Fine-Tuning

In this section, we demonstrate SMORE-DRL's molecular optimization performance against three other DRL methods, as well as its scalability and generalization abilities. For the masker, actor and critic models, encoder weights are not frozen. Additionally, the actor model uses the same head from the pretraining phase. We show that these configurations achieve optimal results in our experiments.

### 4.2.1 Performance Comparison of SMORE-DRL against other DRL Methods for Molecular Optimization

We compare SMORE-DRL's optimization performance with three other MODRL optimization frameworks: (1) DeepFMPOv3D, (2) DeepFMPO and (3) MolDQN. A primary goal of this paper is to present the scalability of SMORE-DRL. However, it is not feasible to conduct large-scale optimization using thousands of lead molecules to compare with the other models, as these benchmarks lack the efficiency for scalability. As described in their papers, DeepFMPOv3D, DeepFMPO and MolDQN optimized a set of 138, 387 and 800 lead molecules, respectively. To facilitate comparison with these methods, a small-scale dataset was utilized. Scalability and generalizability of SMORE-DRL are demonstrated in the following sections. Challenging property values were selected for the optimization task, as noted in Table 1. The results of this comparative study represent the mean scores of three separate runs for all models.

Table 1: Targeted Molecular Properties and Their Maximal Thresholds and True Minimum/Lead Mean Score

| Property | Target Value | True Min/Lead Mean |
|---|---|---|
| ClogP | <3 | -3 (True Min) |
| SAS | <2.5 (2.75 for Testing) | 1 (True Min) |
| BAS (LPA1) | <-6 | -5.27 (Lead Mean) |

All models were trained on a subset of 1,000 lead molecules from the DrugBank database (Wishart et al., 2018) that do not satisfy all properties. SMORE-DRL trained for 70 epochs, during which all lead molecules were optimized over 4 timesteps per epoch. The molecules optimized in the final timestep of the last epoch are used for our comparisons. DeepFMPOv3D and DeepFMPO trained for 1,000 epochs, optimizing random batches of 512 unique molecules per epoch. DeepFMPOv3D performed optimization over 4 timesteps and DeepFMPO used 8 timesteps. In the final epoch, all lead molecules were optimized, and results from this epoch are used for our comparisons. MolDQN was trained for 6,000 epochs. For the first 5,000, a random molecule from the dataset is selected and optimized for 20 timesteps. The final 1,000 epochs focus on optimizing each lead molecule, with the resulting optimized molecules utilized for comparison. To calculate property rewards, all models use the normalization method described in Section 3.3.2. In DeepFMPOv3D and DeepFMPO, a single cumulative reward for all objectives is assigned in the final timestep. For MolDQN, a partial reward excluding BAS is assigned at each intermediate timestep, while a full reward is assigned in the final timestep.

Figure 2 compares each model's learning progression while Table 2 displays the target property percentages achieved by the lead molecules and the optimized outputs of the various models. MolDQN is excluded from Figure 2 as it optimizes one lead molecule per epoch. Figure 8 of Appendix A.7 depicts the property-wise distributions of the final epoch. The optimization capabilities of SMORE-DRL clearly surpass those of the other models. While all other models struggled heavily with the optimization task, SMORE-DRL maintained stability throughout training. Further, it successfully optimized 23.54% of molecules to

meet all properties, while maintaining comparable computation time. The next best model, DeepFMPO, managed only 0.92%. MolDQN had the worst overall performance, failing to produce a single molecule that achieved all properties.

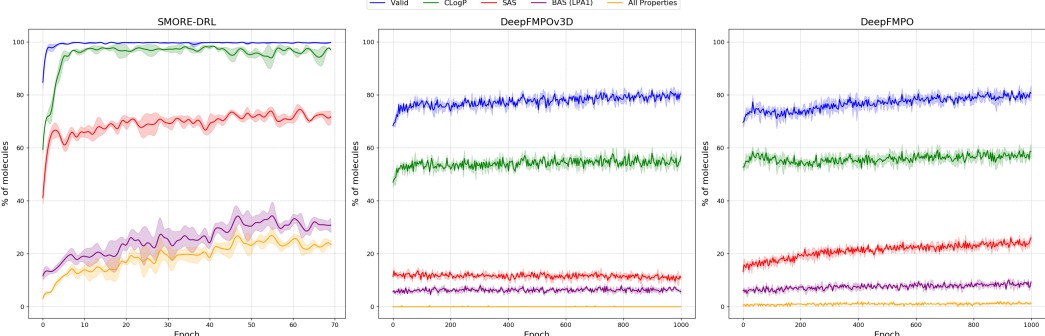

Figure 2: Percentages of valid molecules and those achieving target properties through training for SMORE-DRL, DeepFMPOv3D, and DeepFMPO.

Table 2: Percentage of molecules that satisfy each property from the 1,000 lead molecules and the molecules optimized in the last training epoch by SMORE-DRL, DeepFMPOv3D, DeepFMPO, and MolDQN.

| Property | Lead Molecules | SMORE-DRL | DeepFMPOv3D | DeepFMPO | MolDQN |
|---|---|---|---|---|---|
| Compute Time | - | ~6 hrs | ~4.5 hrs | ~6 hrs | ~5.5 hrs |
| Validity | - | 99.80% (±0.00) | 80.00% (±2.16) | 77.33% (±3.09) | **100% (±0.00)** |
| Novelty | - | 98.52% (±0.05) | 80.01% (±1.85) | 82.99% (±0.27) | **100% (±0.00)** |
| Uniqueness | - | 93.63% (±1.47) | 79.68% (±2.09) | 82.71% (±0.30) | **100% (±0.00)** |
| ClogP | 73.94% | **97.25% (±0.26)** | 54.90% (±0.83) | 61.78% (±2.44) | 87.6% (±0.94) |
| SAS | 38.93% | **71.83% (±2.29)** | 10.96% (±0.60) | 22.86% (±3.82) | 0.70% (±0.29) |
| BAS (LPA1) | 8.65% | **30.85% (±2.77)** | 6.60% (±0.17) | 7.86% (±1.43) | 1.73% (±0.66) |
| All Properties | 0% | **23.54% (±1.56)** | 0.03% (±0.05) | 0.92% (±0.42) | 0% (±0.00) |

### 4.2.2 SCALABILITY OF SMORE-DRL

To examine scalability, SMORE-DRL was trained for 70 epochs using 10,000 molecules-5,000 from the DrugBank database (Wishart et al., 2018) and 5,000 from the Collection of Open Natural Products (COCONUT) database (Sorokina et al., 2021). COCONUT molecules were included to attempt to test the model's robustness, as they are different from those typically encountered during pretraining. All lead molecules were optimized over 4 timesteps per epoch, with those from the final timestep of the last epoch used for comparisons. While the baseline model was run five times, ablation studies were also conducted. These included: (1) without the use of AMTL, (2) with a weight emphasis on the BAS reward (0.25 for ClogP, 0.25 for SAS, and 0.5 for BAS), (3) with the freezing of encoder weights for all agents, (4) with the use of pretraining on the 100-frequency dictionary, and (5) with the use of pretraining on the 1000-frequency dictionary. Figure 3 demonstrates that freezing encoder weights and pretraining on the 100-frequency and 1000-frequency dictionaries significantly impair the model's learning progress. As such, these experiments were limited to a single run of 25 epochs and excluded from further analysis. To evaluate the impact of omitting AMTL and placing greater emphasis on the BAS reward, these model variations were run three times for 70 epochs, while the baseline model ran five timess. Presented results are based on run averages.

As displayed in Figure 3, incorporating AMTL improves training stability and enhances the overall quality of optimized molecules. Findings in Table 3 support this by demonstrating that omitting AMTL significantly impairs most properties, namely uniqueness.

The baseline (SMORE-DRL) model can effectively scale to optimize thousands of lead molecules in a timely manner, even if the molecules are structurally distinct from those used during pretraining. This demonstrates its scalability, efficiency and robustness. Figure 9 (Appendix A.7) exhibits the property-wise distributions, while examples of lead molecules optimized by SMORE-DRL from these experiments are presented in Appendix A.8.

Interestingly, emphasizing the BAS reward did not necessarily produce molecules that were more optimized for BAS compared to the baseline version of SMORE-DRL. A possible explanation for this is that doing so may constrict the model's exploration of the search space, leading it to focus primarily on BAS. This narrow focus may result in the model converging to a local minimum, hindering its ability to discover more optimal solutions in other areas of the search space. A more effective approach would be to implement a dynamic weighting system, initially assigning equal weights to encourage exploration. Over time, these weights could be adjusted to prioritize specific properties.

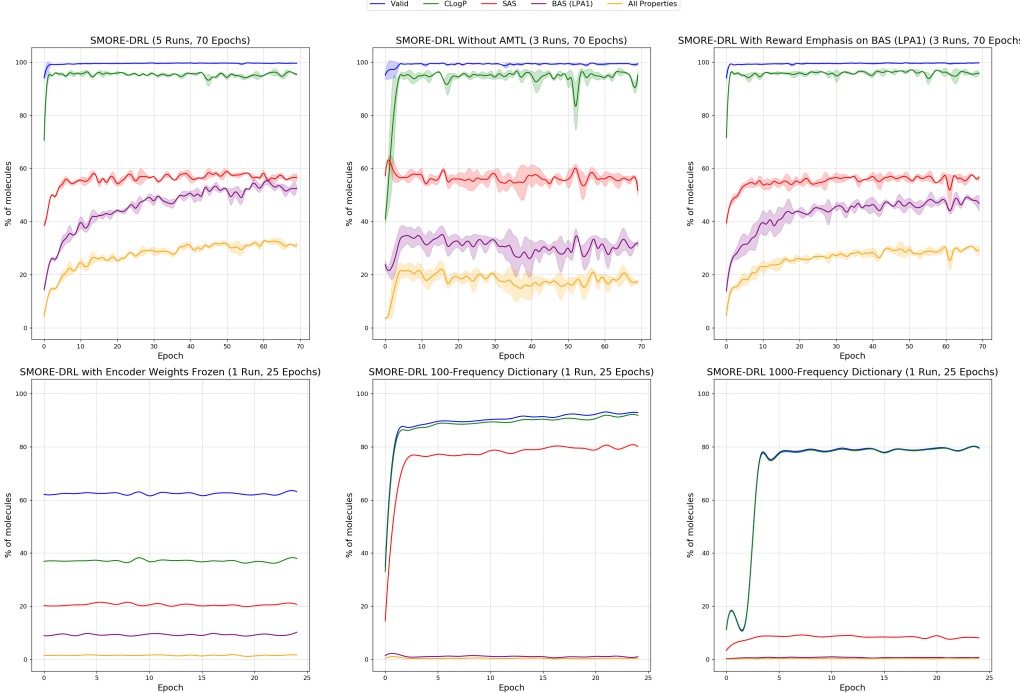

Figure 3: Percentages of valid molecules and those achieving target properties through training for different versions of SMORE-DRL.

Table 3: Percentage of molecules that satisfy each property from the 10,000 lead molecules and the molecules optimized in the last training epoch by SMORE-DRL, SMORE-DRL without AMTL, and SMORE-DRL with a reward emphasis on BAS.

| Property | Lead Molecules | SMORE-DRL | No AMTL | BAS Reward Focus |
|---|---|---|---|---|
| Compute Time | - | ~60 hrs | ~60 hrs | ~60 hrs |
| Validity | - | 99.54% ($\pm$0.08) | 99.33% ($\pm$0.34) | **99.64% ($\pm$0.10)** |
| Novelty | - | 99.98% ($\pm$0.01) | 99.97% ($\pm$0.02) | **99.99% ($\pm$0.00)** |
| Uniqueness | - | 89.31% ($\pm$1.59) | 67.30% ($\pm$3.88) | **90.49% ($\pm$1.91)** |
| ClogP | 62.06% | 94.57% ($\pm$1.35) | 95.04% ($\pm$1.90) | **95.83% ($\pm$0.50)** |
| SAS | 25.64% | **57.08% ($\pm$0.92)** | 51.67% ($\pm$2.17) | 56.74% ($\pm$0.70) |
| BAS (LPA1) | 9.79% | **53.87% ($\pm$2.05)** | 31.71% ($\pm$1.30) | 46.79% ($\pm$2.90) |
| All Properties | 0% | **32.22% ($\pm$1.15)** | 17.20% ($\pm$1.03) | 29.20% ($\pm$1.92) |

### 4.2.3 Generalization Performance of SMORE-DRL

While many MODRL drug design frameworks focus on optimization tasks, their ability to generalize and optimize molecules that they have not encountered before remains unexplored. The weights of the baseline SMORE-DRL model from the scalability experiments were frozen, with their optimization process tested on 40,000 molecules from the COCONUT dataset that differ from those used in the scalability experiments. The following are the average results of the five SMORE-DRL model runs from the fine-tuning phase. To encourage similarity to lead molecules, optimization was restricted to two timesteps. Additionally, the SAS target maximum parameter was increased from 2.5 to 2.75.

SMORE-DRL took 1.25 hours to optimize a test set of 40,000 lead molecules, none of which originally achieved all target properties. 19% of the resulting molecules met all target properties, and all properties were significantly improved (see Table 4). Property-wise distributions are seen in Figure 10 of Appendix A.7, and examples of lead molecule optimized are presented in Appendix A.9.

Table 4: Generalization results – percentage of molecules that satisfy each property from the 40,000 test lead molecule set and the molecules optimized by SMORE-DRL over two timesteps.

| Property | Lead Molecules | SMORE-DRL |
|---|---|---|
| Avg Compute Time | - | ~1.25 hrs |
| Validity | - | 99.44% ($\pm$0.14) |
| Novelty | - | 99.81% ($\pm$0.10) |
| Uniqueness | - | 89.50% ($\pm$0.94) |
| ClogP | 51.22% | 86.01% ($\pm$2.53) |
| SAS | 39.15% | 51.55% ($\pm$1.49) |
| BAS (LPA1) | 16.90% | 38.58% ($\pm$1.62) |
| All Properties | 0% | 19.11% ($\pm$0.55) |

A more detailed discussion of results is found in Appendix A.10.

## 5 Conclusion

In this work, we present SMORE-DRL, a scalable gradient-alignment-based MODRL framework for molecular optimization. A novel hybrid fragment-SMILES representation to depict molecules enables SMORE-DRL to selects and replace fragments in the lead molecules with alternatives from the fragment dictionary, resulting in improved drug candidates. This is achieved by using three agents: a masker, actor and critic, all pretrained on MLM and contrastive learning tasks. SMORE-DRL excelled as a lead molecular optimizer, significantly outperforming other MODRL models while demonstrating scalability. Furthermore, when evaluated on new molecules post fine-tuning, SMORE-DRL effectively generalized its optimization process. The next development of SMORE-DRL will include additional measures to encourage the model to produce molecules that are as effective as those in the current version, but with greater similarity to lead compounds. The implementation of SMORE-DRL is available at `https://anonymous.4open.science/r/SMORE-DRL-F38B`.

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

## A  APPENDIX

### A.1  TRANSFORMER-ENCODER

Transformer-encoder models are made for pretraining on unlabeled data in a bidirectional fashion (Vaswani et al., 2017; Devlin et al., 2019; Shreyashree et al., 2022). To extract features, an embedding layer transforms the input fragment tokens $x = (x_1, x_2, \ldots, x_n)$ into learnable embedding vectors $w = (w_1, w_2, \ldots, w_n)$, with the addition of a sinusoidal positional encoding vectors to reflect sequential location information. This is done using an embedding dictionary $\mathbf{D} \in \mathrm{R}^{\mathrm{V} \times \mathrm{F}}$, where $w_i \in \mathrm{R}^{\mathrm{F}}$, $\mathrm{V}$ is the vocabulary size, and $\mathrm{F}$ is the embedding vector size. As an input feature matrix $\mathbf{Y} \in \mathrm{R}^{\mathrm{N} \times \mathrm{F}}$ is passed through the multi-head self-attention layer, it is linearly transformed into the following $h = 1, 2, \ldots, \mathrm{H}$ matrices: (1) the query matrix $\mathbf{Q}_h = \mathbf{Y}\mathbf{W}_h^{\mathrm{Q}}$, (2) the key matrix $\mathbf{K}_h = \mathbf{Y}\mathbf{W}_h^{\mathrm{K}}$, and (3) the value matrix $\mathbf{V}_h = \mathbf{Y}\mathbf{W}_h^{\mathrm{V}}$, where $\mathbf{W}_h^{\mathrm{Q}}, \mathbf{W}_h^{\mathrm{K}}$, and $\mathbf{W}_h^{\mathrm{V}}$ are model weight matrices. The scaled dot-product attention is then computed for each linear projection, producing the output for a single attention head: $\mathbf{O}_h = \mathrm{softmax}\left(\frac{\mathbf{Q}_h \mathbf{K}_h^{\mathrm{T}}}{\sqrt{d_\mathrm{k}}}\right) \mathbf{V}_h$, where $\sqrt{d_k}$ is a scaling factor. To get the final attention output, all attention heads $\mathbf{O}_1, \mathbf{O}_2, \ldots, \mathbf{O}_{\mathrm{H}}$ are then concatenated and fed into a linear layer. Finally, during pretraining, the attention output is processed by a feed-forward network, referred to as the "pretraining head." This head is typically replaced with task-specific head during the fine-tuning stage (Zhang et al., 2022).

The Masked Language Model (MLM) task, a denoising-based auto-encoding technique, is often used to pretrain encoder models (Devlin et al., 2019). The goal is to reconstruct a noisy token sequence, where some tokens are masked, back to its original form. The model achieves this by using the surrounding visible tokens to build context for predicting the masked tokens (Zhang et al., 2022). More formally, given an input token sequence $x$, a noisy version $\tilde{x}$ is generated by masking a percentage $m$ of its tokens (Wettig et al., 2023). The model's task is to predict on the masked token set $\mathcal{M}$ of $\tilde{x}$ to recover $x$:

$$L(\mathcal{C}) = \mathbb{E}_{x \in \mathcal{C}} \mathbb{E}_{\substack{\mathcal{M} \subset x \\ |\mathcal{M}| = m|x|}} \left[ \sum_{x_i \in \mathcal{M}} \log p\left(x_i \mid \tilde{x}\right) \right]. \tag{7}$$

### A.2  REINFORCEMENT LEARNING

The MDP is defined by the tuple $(\mathcal{S}, \mathcal{A}, \mathcal{P}, \mathcal{R}, \gamma)$, where $\mathcal{S}$ and $\mathcal{A}$ represent the state and action spaces, $\mathcal{P}$ is the state transition probability distribution $\mathcal{P}(s_{t+1}|s_t, a_t)$, $\mathcal{R}$ is the reward distribution $\mathcal{R}(r_t|s_t, a_t)$, and $\gamma$ is the discount factor used to control the trade-off between immediate rewards and future rewards, where $t$ is the current timestep, and $r_t$ is a scalar reward function for $t$ (Al-Jumaily et al., 2023; Graesser & Keng, 2019). The goal of an RL agent is to learn a policy distribution $\pi(a_t|s_t)$ that maximizes long-term cumulative rewards through exploration of the environment over multiple timesteps. This is accomplished by the agent starting at state $s_t$, selecting action $a_t$, receiving reward $r_t$, and transitioning to the new state $s_{t+1}$ (Sutton & Barto, 2018). To assess the value of states and actions with respect to expected long-term returns, two functions are formulated: $V^\pi(s)$, which measures the desirability of $s$: $V^\pi(s) = \mathbb{E}_{s_0 = s, \tau \sim \pi} \left[\sum_{t=0}^{T} \gamma^t r_t\right]$, and $Q^\pi(s, a)$, which measure the desirability of taking action $a$ given state $s$: $Q^\pi(s, a) = \mathbb{E}_{s_0 = s, a_0 = a, \tau \sim \pi} \left[\sum_{t=0}^{T} \gamma^t r_t\right]$ (Graesser & Keng, 2019).

When the RL task entails exploring a vast state or action space, as is often the case in drug design, learning an exact optimal policy or value function can become computationally intractable. Thus, deep reinforcement learning (DRL) is used to approximate policies or value functions (Arulkumaran et al., 2017). The actor-critic framework approximates both and has been leveraged by various drug development frameworks (Al-Jumaily et al., 2023; Goel et al., 2021; Gottipati et al., 2021; Pereira et al., 2021; Popova et al., 2018; Ståhl et al., 2019; Tang et al., 2023; Wang & Zhu, 2024; Yang et al., 2021). The actor model is responsible for learning a parameterized policy $\pi_{\theta_A}$, guided by feedback, known as temporal difference (TD) error from the critic model, which evaluates the actor's actions based on

the state. One approach to this evaluation is by learning the advantage function $A^\pi(s, a) = Q^\pi(s, a) - V^\pi(s)$, which measures the desirability of taking action $a$ compared to alternative actions available from state $s$ (Graesser & Keng, 2019). However, having the critic model learn both $Q^\pi(s, a)$ and $V^\pi(s)$ is computationally expensive. Therefore, in practice, the critic model only learns $V^\pi(s)$ and combines it with reward information from the trajectory to estimate the advantage function:

$$
\begin{aligned}
A^\pi(s_t, a_t) &= Q^\pi(s_t, a_t) - V^\pi(s_t) \\
&\approx r_t + \gamma r_{t+1} + \gamma^2 r_{t+2} + \cdots + \gamma^n r_{t+n} + \gamma^{n+1} \hat{V}^\pi(s_{t+n+1}) - \hat{V}^\pi(s_t).
\end{aligned}
\tag{8}
$$

Thus, the value function is parameterized as $V_{\theta_C}^\pi(s)$ and is updated using loss function:

$$
L_{\mathrm{val}}(\theta_C) = \frac{1}{T} \sum_{t=0}^{T} \left( r_t + \hat{V}_{\theta_C}^\pi(s_{t+1}) - V_{\theta_C}^\pi(s_t) \right)^2,
\tag{9}
$$

while the loss function for the actor is given by:

$$
L_{\mathrm{pol}}(\theta_A) = \frac{1}{T} \sum_{t=0}^{T} \left( -\hat{A}^\pi(s_t, a_t) \log \pi_{\theta_A}(a_t \mid s_t) \right).
\tag{10}
$$

### A.3 AMTL-BASED MODRL ALGORITHM

For the MODRL training, we aim to use the gradient modulation method AMTL (Senushkin et al., 2023) for policy learning. AMTL specifically addresses the multi-task optimization challenges, i.e., gradient dominance and gradient conflicts, by aligning principal components of a gradient matrix. The existence of conflicting or dominating gradients disrupts the stability of the training process and leads to a deterioration in overall performance.

It is acknowledged that the gradient dominance can be measured with a gradient magnitude similarity (Yu et al., 2020), and a cosine distance between vectors can measure the gradient conflicts (Liu et al., 2021). However, the two metrics cannot offer a comprehensive assessment if taken in isolation. One of the key components of AMTL is the proposal of the condition number, a stability criterion that can indicate the presence of both challenges. The value of the condition number is the ratio of the maximum and minimum singular values of the corresponding matrix. Minimizing the condition number of the linear system of gradients, a linear combination of gradients for all objectives, mitigates dominance and conflicts within this system. If we apply singular value decomposition (SVD), we can have

$$
\boldsymbol{G} = \boldsymbol{U}\boldsymbol{\Sigma}\boldsymbol{V}^{\mathrm{T}},
\tag{11}
$$

where $\boldsymbol{\Sigma} = \mathrm{diag}(\sigma_1, \sigma_2, \cdots, \sigma_K)$ and the eigen-values are arranged in decreasing order. One can easily obtain that

$$
\boldsymbol{G}^{\mathrm{T}}\boldsymbol{G} = \boldsymbol{V}\boldsymbol{\Sigma}\boldsymbol{U}^{\mathrm{T}}\boldsymbol{U}\boldsymbol{\Sigma}\boldsymbol{V}^{\mathrm{T}} = \boldsymbol{V}\boldsymbol{\Sigma}\boldsymbol{\Sigma}\boldsymbol{V}^{\mathrm{T}} = \boldsymbol{V}\boldsymbol{\Lambda}\boldsymbol{V}^{\mathrm{T}},
\tag{12}
$$

where $\boldsymbol{\Lambda} = \mathrm{diag}(\lambda_1, \lambda_2, \cdots, \lambda_K)$ and we know that $\sigma_k = \sqrt{\lambda_k}$. Thus, the singular values in the SVD of $\boldsymbol{G}$ correspond to the squared roots of the eigen-values from the eigen-decomposition of the Gram matrix $\boldsymbol{G}^{\mathrm{T}}\boldsymbol{G}$. According to AMTL, a gradient matrix with a minimal condition number (i.e., the singular values are equal to the last positive singular value) can be decomposed as:

$$
\widehat{\boldsymbol{G}} = \boldsymbol{U}\widehat{\boldsymbol{\Sigma}}\boldsymbol{V}^{\mathrm{T}} = \boldsymbol{U}\sigma \boldsymbol{I}\boldsymbol{V}^{\mathrm{T}} = \sigma \boldsymbol{U}\boldsymbol{V}^{\mathrm{T}} = \sigma \boldsymbol{G}\boldsymbol{V}\boldsymbol{\Sigma}^{-1}\boldsymbol{V}^{\mathrm{T}},
\tag{13}
$$

where $\sigma = \sqrt{\lambda_K}$ and $\boldsymbol{U} = \boldsymbol{G}\boldsymbol{V}\boldsymbol{\Sigma}^{-1}$ because of Equation (11), and $\widehat{\boldsymbol{G}}$ is the aligned gradient matrix. A linear combination of the aligned objective-specific gradient vectors using the objective importance would be $\widehat{\boldsymbol{G}}\boldsymbol{\omega} = \sum_{k=1}^{K} \omega_k \widehat{\boldsymbol{g}}_k$. The gist of AMTL is to align the gradient matrix by conducting an SVD to the original gradient matrix and rescaling the singular values to match the smallest singular value. The pseudocode for the MODRL fine-tuning algorithm proposed in this work to align the language model is given in Algorithm 1.

---

**Algorithm 1:** Multi-Objective Deep Reinforcement Learning (MODRL) Pseudocode

---

**Require:** $\pi_0$: original policy; $K$: number of objectives; $\boldsymbol{\omega}$: task importance (all
objectives are deemed equal importance in this work); $\eta$: learning rate;

**1** Let $\pi_\phi = \pi_0$;
**2 foreach** *epoch* **do**
**3**  **foreach** *minibatch* **do**
**4**   **foreach** $k = 1, 2, ..., K$ **do**
**5**    Compute loss $\mathcal{L}_k(\phi)$;
**6**    Compute gradient $\boldsymbol{g}_k = \nabla_\phi \mathcal{L}_k(\phi)$;
**7**   **end**
**8**   Get the gradient matrix $\boldsymbol{G} = \{\boldsymbol{g}_1, ..., \boldsymbol{g}_K\}$; `// playing objective-specific`
`gradient vectors as columns in` $\boldsymbol{G}$
**9**   Compute task space Gram matrix $\boldsymbol{M} \leftarrow \boldsymbol{G}^{\mathrm{T}} \boldsymbol{G}$;
**10**   Get eigen-values and eigen-vectors $(\boldsymbol{\lambda}, \boldsymbol{V}) \leftarrow \mathrm{eigen}(\boldsymbol{M})$;
`// eigen-decomposition such that` $\boldsymbol{M} = \boldsymbol{V}\boldsymbol{\Lambda}\boldsymbol{V}^{\mathrm{T}}$ `where` $\boldsymbol{\Lambda} = \mathrm{diag}(\boldsymbol{\lambda})$
**11**   $\boldsymbol{\Sigma}^{-1} \leftarrow \mathrm{diag}\left(\sqrt{\frac{1}{\lambda_1}}, ..., \sqrt{\frac{1}{\lambda_K}}\right)$;
**12**   Balance transformation $\boldsymbol{B} \leftarrow \sqrt{\lambda_n} \boldsymbol{V} \boldsymbol{\Sigma}^{-1} \boldsymbol{V}^T$;
**13**   Get new aligned gradient matrix $\widehat{\boldsymbol{G}} = \boldsymbol{G}\boldsymbol{B}$; Updated gradient $\nabla\phi = \widehat{\boldsymbol{G}}\boldsymbol{\omega}$;
**14**   Update policy parameter $\phi = \phi - \eta\nabla\phi$;
**15**  **end**
**16 end**
**17** Return policy $\pi_\phi$;

---

### A.4 Fragments-SMILES Hybrid Tokenization Strategy Figures

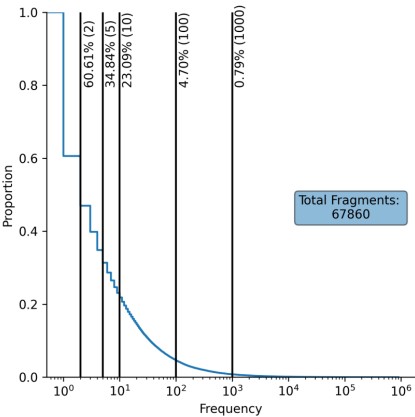

Figure 4: TDC MolGen task dataset (Huang et al., 2021) fragment frequencies.

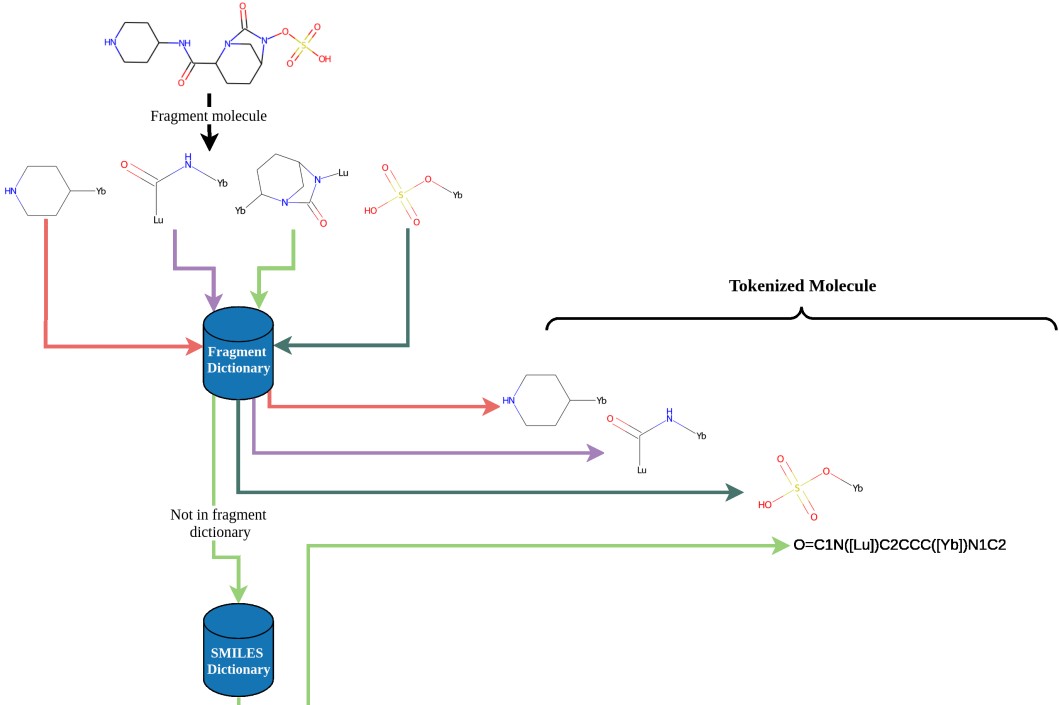

Figure 5: Fragment-SMILES hybrid tokenization strategy.

## A.5 PRETRAINING DIAGRAMS

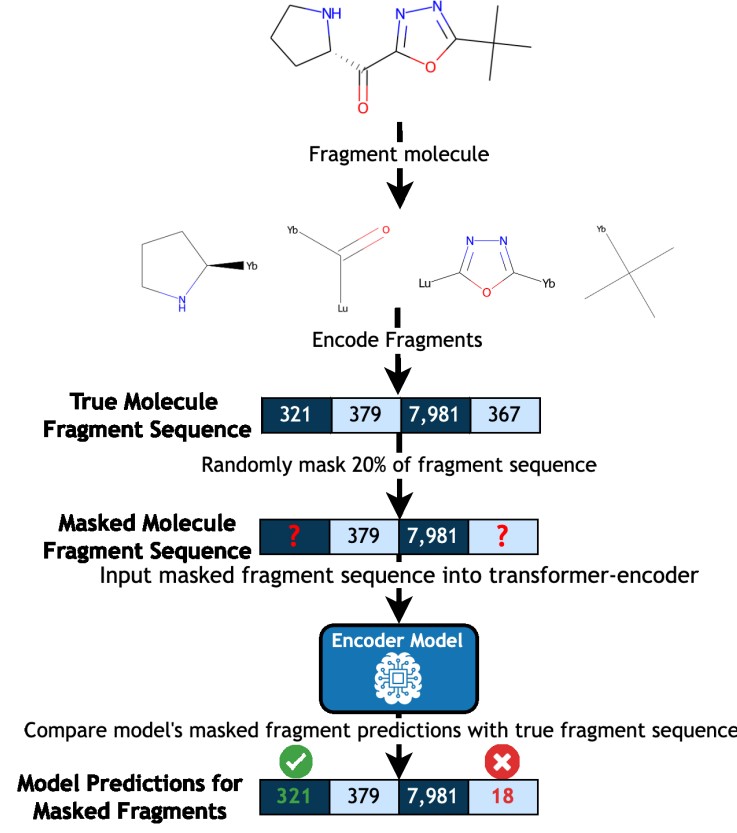

Figure 6: MLM training process for one molecule.

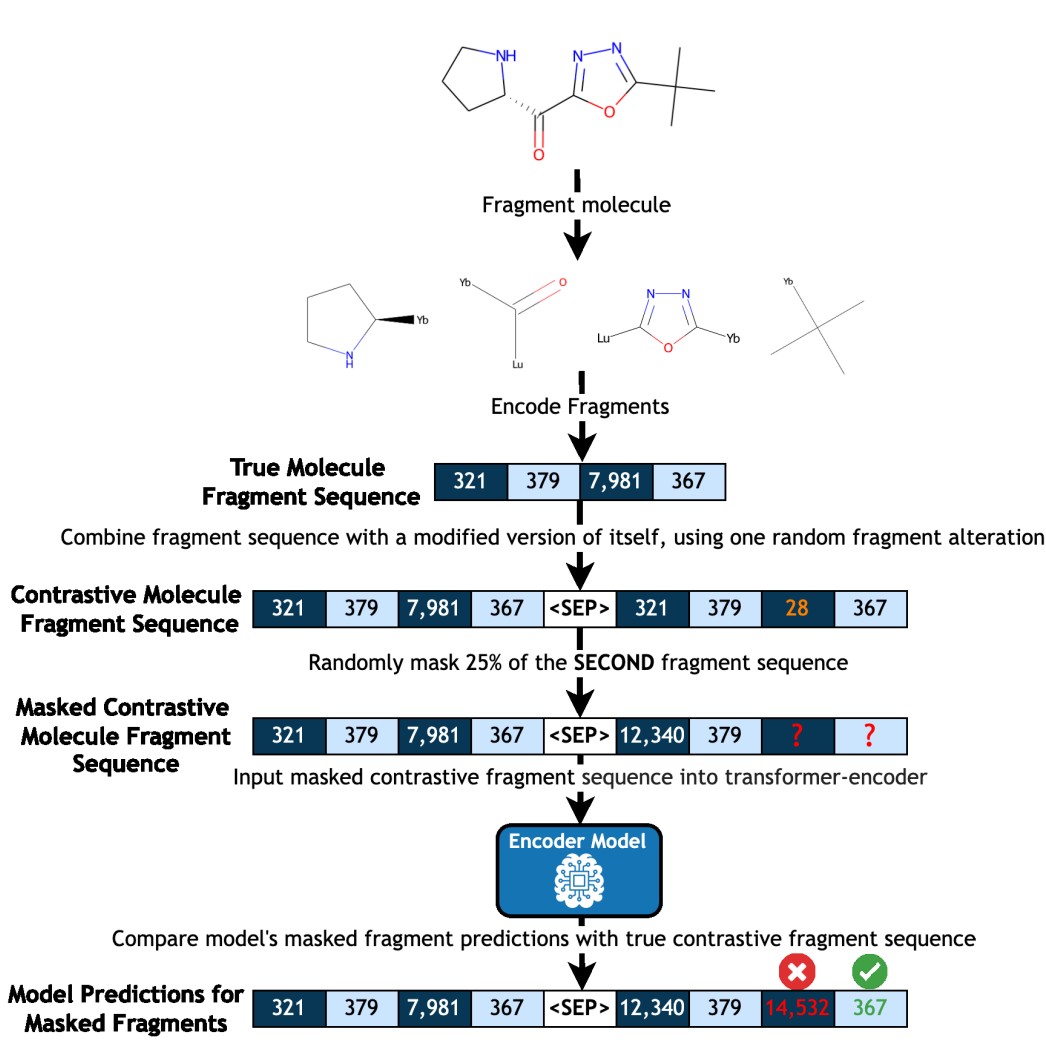

Figure 7: Contrastive learning training process for one molecule.

## A.6 PRETRAINING RESULTS

Table 5: Pretraining results using the 2-Frequency token dictionary on a 4-million molecule dataset from TDC (Huang et al., 2021).

| Epoch | Testing Loss | Testing Accuracy | Compute Time |
|---|---|---|---|
| Epoch 1: MLM | 1.02 | 0.72 | ~8 hrs |
| Epoch 2: MLM | 0.93 | 0.75 | ~8 hrs |
| Epoch 3: Contrastive Learning | 0.37 | 0.90 | ~20 hrs |
| Epoch 4: MLM | 0.87 | 0.78 | ~8 hrs |
| Epoch 5: Contrastive Learning | 1.70 | 0.91 | ~20 hrs |
| Epoch 6: MLM | 0.87 | 0.79 | ~7 hrs |

Table 6: Pretraining results using the 100-Frequency token dictionary on a 4-million molecule dataset from TDC (Huang et al., 2021).

| Epoch | Testing Loss | Testing Accuracy | Compute Time |
|---|---|---|---|
| Epoch 1: MLM | 0.90 | 0.70 | ~3 hrs |
| Epoch 2: MLM | 0.83 | 0.72 | ~3 hrs |
| Epoch 3: Contrastive Learning | 0.29 | 0.89 | ~14 hrs |
| Epoch 4: MLM | 0.80 | 0.73 | ~3 hrs |
| Epoch 5: Contrastive Learning | 0.26 | 0.90 | ~15 hrs |
| Epoch 6: MLM | 0.77 | 0.74 | ~3 hrs |

Table 7: Pretraining results using the 1000-Frequency token dictionary on a 4-million molecule dataset from TDC (Huang et al., 2021).

| Epoch | Testing Loss | Testing Accuracy | Compute Time |
|---|---|---|---|
| Epoch 1: MLM | 0.88 | 0.70 | ~3 hrs |
| Epoch 2: MLM | 0.81 | 0.72 | ~3 hrs |
| Epoch 3: Contrastive Learning | 0.24 | 0.90 | ~15 hrs |
| Epoch 4: MLM | 0.78 | 0.73 | ~3 hrs |
| Epoch 5: Contrastive Learning | 0.22 | 0.91 | ~15 hrs |
| Epoch 6: MLM | 0.76 | 0.74 | ~3 hrs |

## A.7 PROPERTY-WISE DENSITY PLOTS FOR THE COMPARATIVE AND SCALABILITY STUDIES

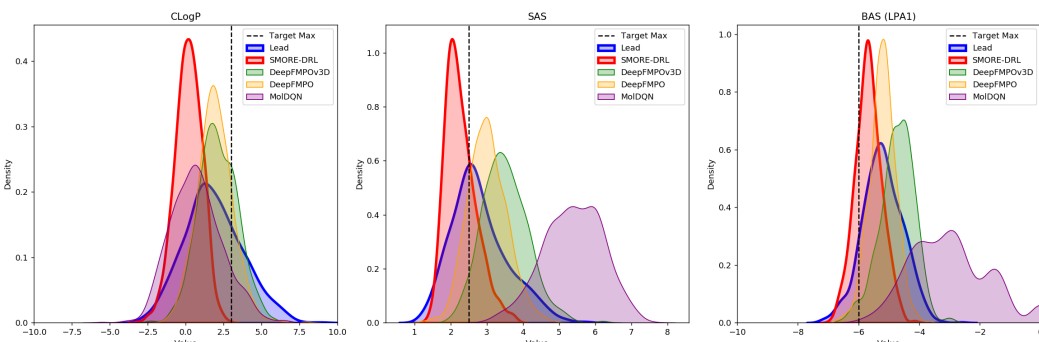

Figure 8: Property-wise comparisons between the lead molecules (blue) and the molecules optimized in final epoch by SMORE-DRL (red), DeepFMPOv3D (Bolcato et al., 2022) (green), DeepFMPO (Ståhl et al., 2019) (yellow), and MolDQN (Zhou et al., 2019) (purple). All objectives are to be minimized and the targeted maximums are indicated by the black dashed line.

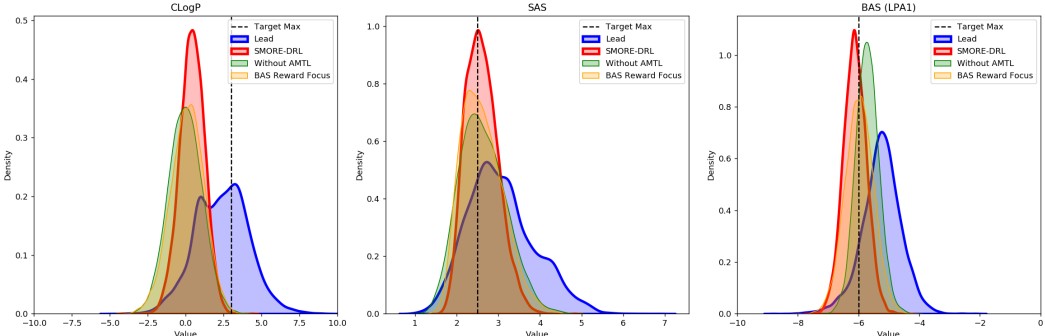

Figure 9: Property-wise comparisons between the lead molecules (blue) and the molecules optimized in final epoch by SMORE-DRL (red), SMORE-DRL without AMTL (green), and SMORE-DRL with a reward emphasis on BAS (yellow). All objectives are to be minimized and the targeted maximums are indicated by the black dashed line.

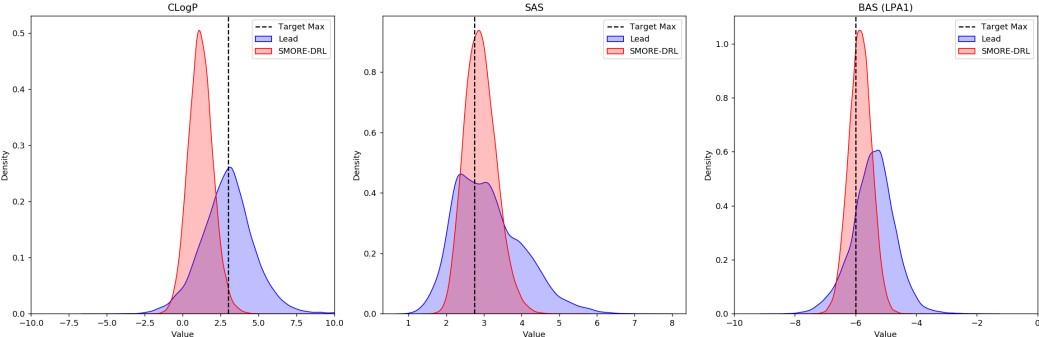

Figure 10: Generalization results – property-wise comparisons between the test lead molecules (blue) and the molecules optimized by SMORE-DRL (red). All objectives are to be minimized and the targeted maximums are indicated by the black dashed line.

## A.8 Visualizations of SMORE-DRL's Molecular Optimization During Scalability Experiments

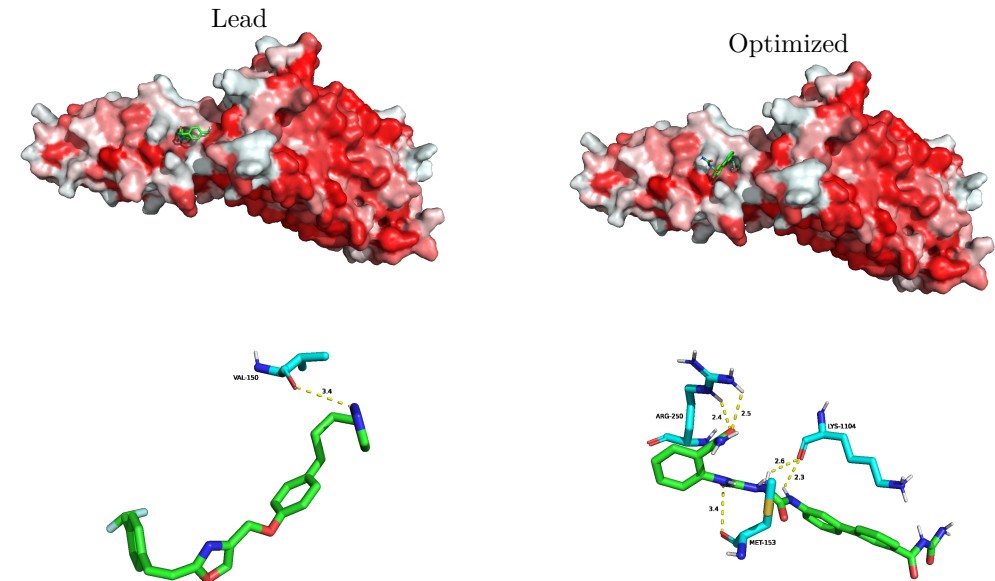

Figure 11: Binding visualization of a lead molecule (`FC(F)(F)c1ccc(C=Cc2nc(COc3ccc(CCCn4ccnn4)cc3)co2)cc1`, ClogP = 6.05, SAS = 2.73, BAS = -4.7) and SMORE-DRL's optimized version (`NC(=O)NC(=O)c1cccc(-c2cccc(NC(=O)NNC(=O)Nc3ccccc3C(N)=O)c2)c1`, ClogP = 2.11, SAS = 2.31, BAS = -9.0) from the scalability experiments.

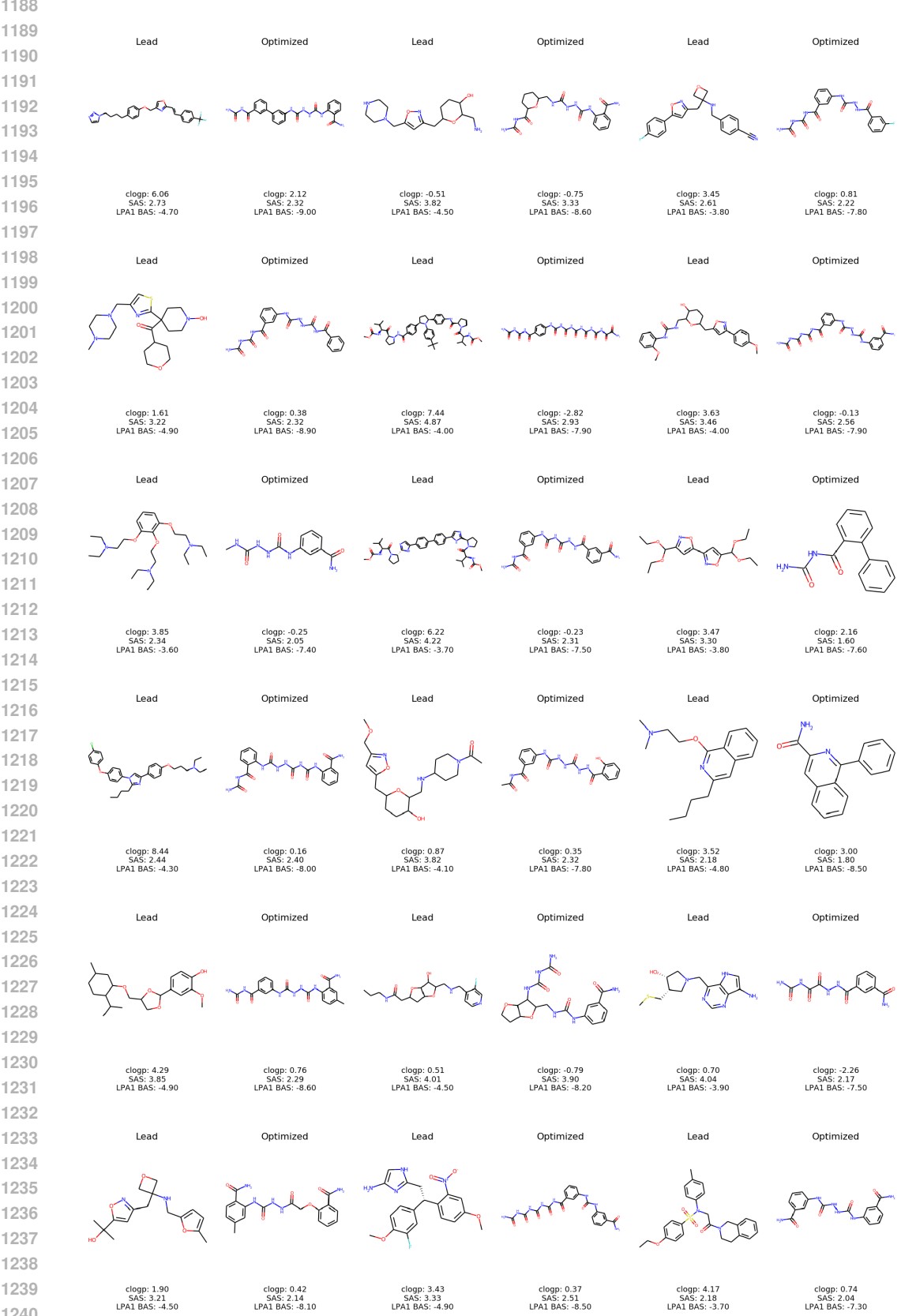

Figure 12: Lead molecules optimized by SMORE-DRL from the scalability experiments.

## A.9 Visualizations of SMORE-DRL's Molecular Optimization During Generalization Experiments

Lead
Optimized

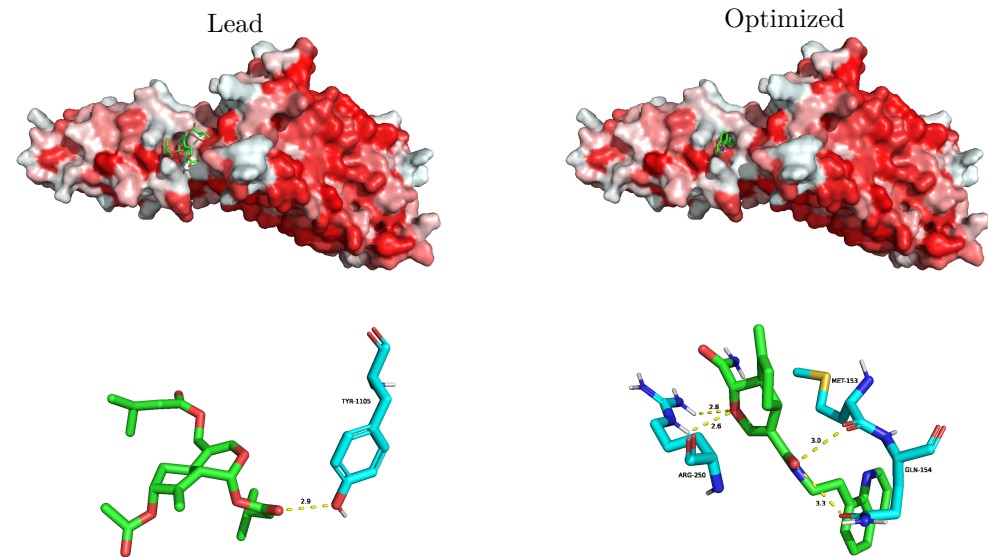

Figure 13: Binding visualization of a lead molecule (`C=C1C(OC(C)=O)CC2C(COC(=O)CC(C)C)=COC(OC(=O)CC(C)C)C12`, ClogP = 3.52, SAS = 4.33, BAS = -3.8) and SMORE-DRL's optimized version (`C=C1CCC2C(C(=O)NCCc3cccc4cccnc34)=COC(C(N)=O)C12`, ClogP = 2.24, SAS = 3.88, BAS = -8.2) from the generalization experiments.

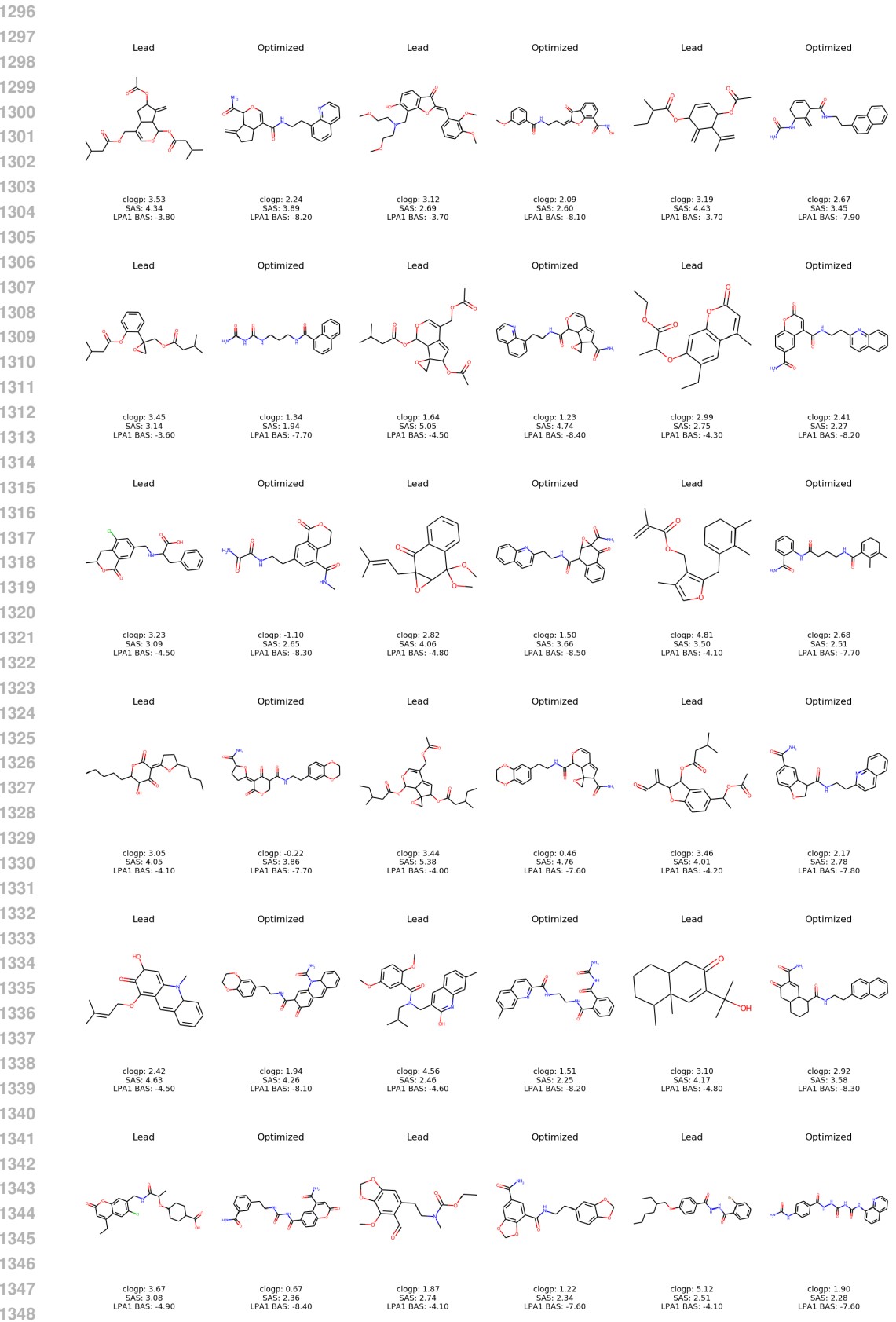

Figure 14: Lead molecules optimized by SMORE-DRL from the generalization experiments.

A.10 Discussion

In this paper, we introduce SMORE-DRL, a novel transformer-based MODRL model for molecular optimization. Three sets of experiments were conducted to evaluate the model's performance: (1) a comparative study against DeepFMPO, DeepFMPOv3D, and MolDQN, three MODRL molecular optimization models, tasked with optimizing 1,000 lead molecules, (2) a scalability study, where the model was tasked with optimizing 10,000 lead molecules, and (3) a generalization study to assess how well the model, after training in the scalability study, can optimize 40,000 lead molecules in a test scenario.

SMORE-DRL demonstrated outstanding performance in all experiments. In the comparative study, it significantly outperformed all other models. In the scalability study, SMORE-DRL performed efficiently, optimizing a set of lead molecules that did not achieve all properties so that one third of produced molecules satisfied all properties. Additionally, SMORE-DRL's robustness allowed it to successfully generalize its optimization approach to unseen molecules. With just two modification steps, it improved the lead molecules from 0% to 19% achieving all target properties. The inclusion of AMTL has proven to be a vital component of SMORE-DRL, enhancing training stability and improving the overall performance.

As discussed, the primary objective of molecular optimization is developing a novel molecule similar to a lead molecule, aiming to have both molecules exhibit comparable qualities. As such, the progression of SMORE-DRL's optimized molecules were analyzed by comparing their similarity to lead molecules and their corresponding rewards across all timesteps. Figure 15 depicts the average similarity and reward for each of the four optimization timesteps performed on 1,000 molecules during the scalability study. To measure similarity, we utilize the method described in DeepFMPO (Ståhl et al., 2019), which employs a combination of maximum common substructure Tanimoto similarity and Levenshtein distance. A similarity score greater than or equal to 0.7 indicates high similarity, while a score between 0.5 and 0.7 is considered medium similarity (Loeffler et al., 2024). While SMORE-DRL does not achieve high similarity, it still presents strong results. As seen in Figure 15, there is an inverse correlation between average similarity and average sum of rewards across all objectives, where as similarity decreases, reward increases. This represents a trade-off: restricting the optimization process to minimal modifications of a lead molecule may result in high similarity, but will likely restrict exploration and hinder the development of superior candidates. Nonetheless, the next iteration of SMORE-DRL should balance exploration with maintaining similarity to lead molecules, aiming to generate high-quality compounds without sacrificing similarity. One possible approach involves incorporation of a dynamic similarity component into the reward function, allowing for exploration in the initial training epochs while penalizing molecules with low similarity to lead molecules in the later epochs.

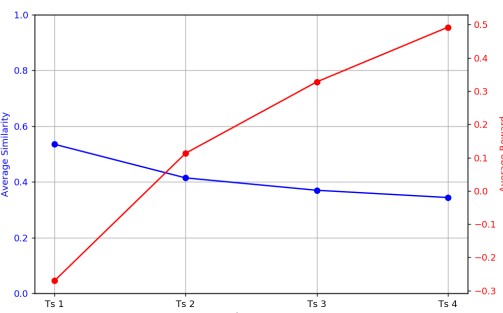

Figure 15: An analysis of: (1) the average similarity to lead molecules and (2) the average sum of rewards across all properties over four optimization timesteps for 1,000 molecules optimized by SMORE-DRL.

