# OpenReview forum: "SMORE-DRL: Scalable Multi-Objective Robust and Efficient Deep Reinforcement Learning for Molecular Optimization"
_ICLR.cc/2025/Conference — ICLR 2025 Conference Withdrawn Submission_

### Official Review · Reviewer_YepM · 2024-10-26

**Soundness:** 3
**Presentation:** 3
**Contribution:** 3
**Rating:** 5
**Confidence:** 3

**Summary:**

This paper proposed a transformer-based multi-objective DRL architecture for the optimization of molecules across multiple pharmacological properties, including binding affinity to a cancer protein target. Specifically, The model consists of a pre-trained BERT model on SMILES string and a multi-objective actor-critic framework for fine-tuning. The authors claim following novel contributions: First, they prepared the data into a Fragments-SMILES Hybrid manner in order to reduce chemical search space and fine-tune time. Second, their actor-critic framework composes of three models: a masker, an actor, and a critic, and the masker is used for the MLM task. Their experiment shows that the model has more satisfied modified molecules than compared baselines across multiple molecule properties.

**Strengths:**

1. This paper proposed a novel Transformer-DRL framework for molecule modification task and achieve good performance.
2. The idea of reducing chemical search space and embed MLM task into DRL framework is great.
3. It uses an anonymous code base for reviewer to check with code

**Weaknesses:**

1. The binding affinity score threshold is set to -6, which does not represent a very good drug to my knowledge. Could you explain why the threshold is picked?
2. From the figure 8 and 9 we could see that model with BAS focus reward does not has more BAS-qualified molecules than the original model. Also, the original model has more BAS-qualified molecules in figure 8 than figure 9. Could you explain these differences? And it would be great to use difference color for each difference models/variants.

**Questions:**

The questions were asked in weaknesses section

---

### Official Review · Reviewer_Ykt8 · 2024-11-01

**Soundness:** 2
**Presentation:** 1
**Contribution:** 1
**Rating:** 3
**Confidence:** 3

**Summary:**

This paper introduces SMORE-DRL, a molecular optimization method built on multi-objective reinforcement learning. SMORE-DRL incorporates a novel tokenization approach that represents molecules as a hybrid of fragments and SMILES, which is used for training a transformer model. Additionally, SMORE-DRL employs a pre-training strategy based on masked language modeling and contrastive learning. In experiments, the authors compare SMORE-DRL with other deep reinforcement learning approaches for molecular optimization, including DeepFMPOv3D, DeepFMP, and MolDQN.

**Strengths:**

- Tackles an important research area: multi-objective molecular optimization.
- Figures 5, 6, and 7 in the Appendix are helpful for understanding the methods.
- SMORE-DRL's performance against baselines is promising, as shown in Table 2.

**Weaknesses:**

- My main concern is that the methodology section lacks clear structure, which limits the overall impact of the paper. For example:
    - Explain how the MDP is constructed before introducing the agent details, as this current organization hinders comprehension.
    - In Equation 4, Rt,k​ is introduced without a clear explanation of the reward’s meaning beforehand.
    - Variables in Equation 2, such as the advantage function A_k​, are introduced without definition.
    - In lines 315–317, the phrase "output of the reward function" is unclear. Are you referring to a multi-objective reward here?
- Randomly switching a fragment could significantly alter molecular properties. How does the method account for these potentially drastic changes?
- Is it accurate to describe the approach as "contrastive learning"? There seems to be no explicit component for repelling representations of negative samples, which is typically characteristic of contrastive learning.
- The font size in Figures 2 and 3 is too small, making them difficult to read. Increasing the font size would improve accessibility.

**Questions:**

See Weaknesses Section

---

### Official Review · Reviewer_3uNg · 2024-11-01

**Soundness:** 2
**Presentation:** 1
**Contribution:** 2
**Rating:** 3
**Confidence:** 4

**Summary:**

This paper presents a deep reinforcement learning framework tailored for multi-objective molecular optimization, addressing challenges in efficiency, scalability, and generalization. By leveraging a hybrid fragment-SMILES tokenization strategy and a novel gradient alignment approach in multi-objective optimization, the proposed model, SMORE-DRL, demonstrates superior performance in optimizing molecular properties such as binding affinity, synthetic accessibility, and partition coefficient. The authors conduct extensive experiments to benchmark SMORE-DRL against other DRL models, showing favorable outcomes in optimization quality, stability, and generalization on unseen molecules.

**Strengths:**

This work tackles a fundamental problem in AI-driven drug design: molecular optimization, which involves enhancing desirable molecular properties while maintaining structural similarity to a lead compound. SMORE-DRL introduces an innovative hybrid fragment-SMILES representation, effectively capturing diverse molecular structures and enabling efficient learning. The paper also includes a well-structured set of experiments to evaluate the model’s performance, showcasing its scalability and stability across large datasets.

**Weaknesses:**

1. **Lack of Similarity Evaluation:** The primary goal of molecular optimization is to generate molecules with enhanced properties that closely resemble the lead molecule. However, the presented experiments do not evaluate similarity to lead molecules, which is a critical metric for molecular optimization. Without similarity assessments, the experimental results are less meaningful and do not fully support claims regarding molecular optimization.

2. **Formatting Issues:** The paper’s formatting is inconsistent with the official template, with the font size in Figures 1, 2, and 3 being too small to read. This issue detracts from the paper’s readability and overall presentation.

3. **Missing Related Works and Baselines:** Important transformer-based methods for molecular optimization are omitted from the discussion. Including recent competitive models, such as Irwin, et al. (2022), He, et al. (2022), and Loeffler, et al. (2024), as baseline comparisons would improve the experimental rigor and relevance.

   [1] Irwin, Ross, et al. "Chemformer: a pre-trained transformer for computational chemistry." Machine Learning: Science and Technology 3.1 (2022): 015022.

   [2] He, Jiazhen, et al. "Transformer-based molecular optimization beyond matched molecular pairs." Journal of cheminformatics 14.1 (2022): 18.

   [3] Loeffler, Hannes H., et al. "Reinvent 4: Modern AI–driven generative molecule design." Journal of Cheminformatics 16.1 (2024): 20.

4. **Limited Target Evaluation:** Section 4.2.1 evaluates the model on only one protein target, limiting the conclusions that can be drawn about SMORE-DRL’s general effectiveness across diverse targets.

5. **Scalability Contribution Questioned:** While Section 4.2.2 emphasizes SMORE-DRL’s scalability, the contribution appears limited to parallel optimization of multiple lead molecules, which is feasible for other transformer-based models as well and may not represent a significant technical advancement.

**Questions:**

1. Could the authors clarify the specific GPUs used for the experiments, given the reported computation time?
2. In line 305, how is the "partial reward" assigned to incomplete molecules, and what rationale supports this reward structure?
3. All SMILES strings are canonicalized for pretraining; how does the model ensure it does not generate non-canonicalized SMILES? Given that SMILES randomization is a common strategy, why was it not employed here?

---

### Official Review · Reviewer_2Po6 · 2024-11-03

**Soundness:** 1
**Presentation:** 1
**Contribution:** 1
**Rating:** 3
**Confidence:** 4

**Summary:**

In this paper, the authors propose a multi-objective molecule optimization framework based on reinforcement learning to optimize SMILES strings of molecules for better properties. To achieve this, the framework first pre-trains a transformer-encoder for SMILES strings with two tasks: masking and contrastive learning. This pre-trained encoder is then utilized for fine-tuning three models:  a masker model to mask tokens within SMILES strings, an actor model to replace tokens for better SMILES strings, and a critic model to evaluate the expected returns of properties. Experimental results show that the proposed method can outperform baselines.

**Strengths:**

* The method leverages a gradient alignment technique proposed in AMTL to mitigate conflicting and dominating gradients when optimizing toward multiple properties.
* The method designs a reinforcement learning algorithm to optimize molecules.
* The method outperforms baselines in optimizing molecules that satisfy all property requirements.

**Weaknesses:**

* The paper does not include state-of-the-art baselines for multi-objective molecule optimization. Therefore, the experimental results cannot demonstrate the superiority of the proposed method over state-of-the-art multi-objective molecule optimization methods.
* The experimental evaluation cannot support the method’s superior scalability and generalization capabilities over current methods. These capabilities are only analyzed through comparisons with model variants instead of against state-of-the-art baselines.
* The paper is not written well. The notations are very confusing, and the method section does not clearly show the novelty and the contributions of this paper.
* The paper lacks sufficient technical novelty for this conference. Previous methods [1,2] also explored Pareto-based multi-objective molecule optimization and need to be discussed. The molecule tokenization strategy, which simply combines fragment with atom tokens together, based on the reviewer’s opinion, also does not contribute to the technical novelty.

[1] Zhu, Yiheng, et al. "Sample-efficient multi-objective molecular optimization with gflownets." Advances in Neural Information Processing Systems 36 (2024).
[2] Xia, Xin, et al. "Evolutionary Multiobjective Molecule Optimization in an Implicit Chemical Space." Journal of Chemical Information and Modeling 64.13 (2024): 5161-5174.

**Questions:**

* It is strongly suggested to include state-of-the-art molecule optimization baselines. The current baselines are not sufficient to demonstrate the superiority of the proposed method. The authors can use some benchmark papers like [3] as references.
* It is strongly suggested to include other baselines when analyzing the scalability and generalization ability. The reviewer also wonders which design of the proposed method can contribute to its better capabilities compared to baselines.
* It is suggested to dramatically improve the introduction and method section and clearly explain the motivation and benefits of each design choice. For example, it would be helpful to clarify why the pre-trained transformer encoder is necessary or how it contributes to better performance.

[3]  Gao, Wenhao, et al. "Sample efficiency matters: a benchmark for practical molecular optimization." Advances in neural information processing systems 35 (2022): 21342-21357.

---

### Note · Authors · 2024-11-25

**Comment:**

Dear ICLR Committee,

We thank all of our reviewers for their thoughtful and constructive feedback. Your insights have provided us with valuable guidance to improve our work. However, addressing this feedback requires conducting additional experiments that would take longer than the rebuttal period allows. Therefore, we have decided to withdraw our paper at this time. We greatly appreciate the reviewers' time and effort in evaluating our submission and will use the feedback to guide future improvements to this work.

Kind regards,
Aws Al Jumaily

**Withdrawal Confirmation:**

I have read and agree with the venue's withdrawal policy on behalf of myself and my co-authors.